

# Scaling laws for Haralick texture features of linear gradients

Sorinel A. Oprisan and Ana Oprisan

Physics and Astronomy, College of Charleston, Charleston, SC, United States

## ABSTRACT

This study presents a novel analytical framework for understanding the relationship between the image gradients and the symmetries of the Gray Level Co-occurrence Matrix (GLCM). Analytical expression for four key features–sum average (SA), sum variance (SV), difference variance (DV), and entropy–were derived to capture their dependence on image's gray-level quantization ($N_g$), the gradient magnitude ($\nabla$), and the displacement vector (d) through the corresponding GLCM. Scaling laws obtained from the exact analytical dependencies of Haralick features on $N_g$, $\nabla$ and |d| show that SA and DV scale linearly with $N_g$, SV scales quadratically, and entropy follows a logarithmic trend. The scaling laws allow a consistent derivation of normalization factors that make Haralick features independent of the quantization scheme $N_g$. Numerical simulations using synthetic one-dimensional gradients validated our theoretical predictions. This theoretical framework establishes a foundation for consistent derivation of analytic expressions and scaling laws for Haralick features. Such an approach would streamline texture analysis across datasets and imaging modalities, enhancing the portability and interpretability of Haralick features in machine learning and medical imaging applications.

## INTRODUCTION

Researching image texture presents a fundamental challenge: it requires a universally accepted definition. Texture can be perceived through tactile means *Manjunath & Ma (1996)* and optical methods (*Tuceryan & Jain, 1999*). Humans recognize texture in images (*Papathomas, Kashi & Gorea, 1997*; *Aviram & Rotman, 2000*; *Jagadeesh & Gardner, 2022*), distinguishing it by attributes such as coarseness and roughness. The human visual system relies on local contrast ratios and intensity differences, rather than absolute pixel intensity values, to interpret image patterns, such as intensity gradients (*Werner, 1935*; *Land & McCann, 1971*; *Attneave, 1954*; *Barten, 1999*). In non-human primates, neurons selectively respond to surface luminance gradients and utilize linear shading gradients to infer three-dimensional (3D) structure (*Hanazawa & Komatsu, 2001*). While previous experimental findings established that the primate visual cortex prioritizes luminance gradients over absolute luminosity as a key visual feature for pattern classification (*Correani, Scott-Samuel & Leonards, 2006*; *Keil, 2007*), more recent research has demonstrated that image gradients also facilitate the neural encoding of 3D representations of textured objects (*Gomez & Neumann, 2016*).

Corresponding author
Sorinel A. Oprisan,
oprisans@cofc.edu

Furthermore, MRI studies in humans have shown that luminance gradients along the vertical axis of an image elicit stronger neural responses in scene-selective brain regions compared to horizontal gradients (*Cheng, Chen & Dilks, 2023*). This directional selectivity suggests that the human brain assigns different levels of importance to intensity gradients depending on their orientation within natural scenes. Experimental evidence also suggests that vertical intensity gradients are processed by distinct neural pathways in the early visual cortex than those used for gradients in other orientations (*Vaziri et al., 2014*).

Computer applications have leveraged human visual perception by incorporating gradients as fundamental visual features to enhance the informational content of images. For instance, geographic information system (GIS) tools utilize color gradients to represent variations in elevation and population density (*DeMers, 2008*). In image processing, gradients serve as essential components for various tasks, including edge detection (*Canny, 1986*), to correct different lighting or camera properties (*Marchand, 2007*), and distinguishing between digital camera images and scanned images (*Mettripun & Amornraksa, 2014*). Additionally, reducing gradient magnitudes at transitions within mosaic images helps create visually cohesive scenes, which human observers perceive as single, unified images (*Perez, Gangnet & Blake, 2003*).

Natural-scene images depict nature-made objects, such as landscapes, animals, and plants. At the initial stage of an image processing pipeline, basic image enhancement tasks must make assumptions about the image through interpolation methods like smoothing and filtering or model fitting techniques such as Bayesian inference. Although prior knowledge is essential for image processing, it can also introduces bias by favoring expected outcomes. Spectral priors do not directly encode information about an image's specific properties but instead influence its histogram (the spectrum). Many image features, including color and texture, can be derived from image gradients or spectral priors, as they exhibit remarkable invariance across images (*Long & Purves, 2003*; *Tward, 2021*; *Dresp-Langley & Reeves, 2024*). Each pixel in a gradient image contains two values corresponding to the gradient components at that location. The gradient distribution represents these values' histogram or probability distribution across all pixels or multiple images. This study focused on one-dimensional gradients in two-dimensional images to explore how Haralick statistical features relate to image gradients. Significant discrepancies exist between human and machine vision in classifying the same textures (*Tamura, Mori & Yamawaki, 1978*). Efforts to enhance machine-based texture recognition have included detailed models of human visual perception of luminance differences (*Chan, Golub & Mulet, 1970*; *Miao & Shaohui, 2017*) and techniques that focus on grouping similar image regions (*Rosenfeld & Kak, 1982*) or analyzing semi-repetitive pixel arrangements in natural scenes (*Pratt, 1978*, *2006*).

Computer vision and "big data" efficient algorithms driven by machine learning (ML) and Artificial Intelligence (AI) rapidly expanded into the medical imaging field in healthcare. Despite its significance, over 97% of recorded medical images remain unused due to inadequate feature extraction and classification methods (*Murphy, 2019*). With the emergence of ML and AIs, several automated systems for medical image analysis have been developed. These include tools for bone age estimation (*Kim et al., 2017*), detection of

pulmonary tuberculosis and lung nodules (*Hwang et al., 2018*; *Singh et al., 2018*), and AI-based lobe segmentation in CT images (*Fischer et al., 2020*). Texture analysis is crucial in such applications, including diagnosing microcalcifications in breast tissue (*Karahaliou et al., 2007*) and detecting cancer from ultrasound images of various organs (*Faust et al., 2018*).

Texture analysis has been applied to improve the quality of life for individuals with visual impairments. For example, it has enhanced handwriting digit identification accuracy (*Sanchez Sanchez et al., 2024*) and improved the performance of classification algorithm (*Alshehri et al., 2024*). In nondestructive material testing, texture analysis helps characterize changes in microstructure caused by mechanical, thermal, and operational stresses. By analyzing microstructural features, researchers gain a deeper understanding of bulk material properties and their macroscopic mechanical behavior. Microstructure texture classification has been widely used in metallurgical studies, based on second-order statistical features such as Haralick features (*Haralick, Shanmugam & Dinstein, 1973*; *Haralick, 1979*). Applications include identifying constituent metallurgical phases in steel microstructures (*Naik, Sajid & Kiran, 2019*), assessing surface hardening during cooling (*Fuchs, 2005*), detecting phase transitions in two-phase steel systems (*Liu, 2014*), and analyzing the effects of tempering parameters on steel microstructure (*Dutta et al., 2014*). Additionally, texture analysis has been utilized to quantify corrosion in steam piping systems (*Fajardo et al., 2022*). In soft condensed matter, texture classification has been used for identifying phase transitions in polymers and liquid crystals (*Pieprzyk et al., 2022*; *Sastry et al., 2012*) and measuring shear modulus, failure temperature, and zero shear viscosity, in polymeric colloids (*Xu et al., 2024*).

Texture-based image analysis often utilizes advanced statistical methods, such as discriminative binary and ternary pattern features (*Midya et al., 2017*), wavelet-based techniques (*Wan & Zhou, 2010*; *Karahaliou et al., 2007*), and matrix-based approaches such as gray-level run length (*Raghesh Krishnan & Sudhakar, 2013*), autocovariance (*Huang, Lin & Chen, 2005*), and spatial gray-level dependence matrices (*Kyriacou et al., 1997*; *Pavlopoulos et al., 2000*).

One widely used approach to texture analysis is the Gray Level Co-occurrence Matrix (GLCM), a statistical method that captures spatial relationships between pixel intensities (*Oprisan & Oprisan, 2023*). GLCM, which belongs to second-order statistical methods (*Humeau-Heurtier, 2019*), quantifies occurrences of pixel pairs that exhibit specific spatial relationships. *Haralick, Shanmugam & Dinstein (1973)*, *Haralick (1979)* identified 14 texture features derived from GLCM; however, many have been critiqued for redundancy (*Conners & Harlow, 1980*) and computational complexity. Advanced methods, including higher-order statistics and fractal dimensions (*Pavlopoulos et al., 2000*; *Kyriacou et al., 1997*), have further enriched the field but remain limited in practical application due to high computational demands.

The primary objective of this study is to derive analytical expressions for the GLCM and its related features, in order to better understand how they depend on gray-level quantization ($N_g$), image gradient magnitude ($\nabla$), and displacement vector ($d$). The secondary objective is to use these newly derived expressions, particularly those from the

GLCM of linear gradients, to establish scaling laws that govern the dependence of Haralick features on $N_g$, $\nabla$, and $d$. These scaling laws will help determine the asymptotic behavior of Haralick features and identify data-driven normalization factors, ensuring that results remain independent of the image quantization scheme. Previous studies primarily relied on empirical methods to estimate normalization factors that could make Haralick features invariant to the number of gray levels ($N_g$). For instance, *Clausi (2002)* proposed normalizing gray-level intensities by the total number of gray levels in the GLCM, but applied this only to two features—inverse difference and inverse difference moment. Similarly, *Shafiq-ul Hassan et al. (2017*, *2018)* aimed to enhance the reproducibility of MRI-based Haralick features across different voxel volumes and scanner models (Philips, Siemens, and GE models). However, their empirical approach identified only two reproducible GLCM-based features, and they noted that "for some features, their relationship with gray levels appeared to be random, therefore, no normalizing factor could be identified" (*Shafiq-ul Hassan et al., 2017*). *Lofstedt et al. (2019)* also investigated methods to reduce the sensitivity of Haralick features to image size, noise levels, and different quantization schemes. Their approach involved normalizing each gray level by $N_g$ and additional empirical normalization factors, effectively transforming the GLCM into an equivalent normalized Riemann sum. While this normalization improved consistency for many texture features, it did not work universally, although "most of the modified texture features quickly approach a limit." This study introduces a systematic methodology for deriving scaling laws that explain how Haralick features evolve with changes in the number of gray levels ($N_g$). By establishing these scaling laws analytically, we aim to provide a more rigorous foundation for normalization strategies, reducing the reliance on empirical estimations.

This study demonstrates the derivation methodology for feature dependencies on $N_g$, $d$, and $\nabla$ for four Haralick features: sum average (SA), sum variance (SV), difference average (DA), and entropy. We chose these four Haralick features because they have received significantly less attention than those based directly on calculating various moments of the GLCM. Examples include Second Angular Moment or Energy $f_1$ (over 19,300 publications in Google Scholar), Contrast $f_2$ (22,500 publications), Correlation $f_3$ (21,000 publications), Sum of Squares Variance $f_4$ (20,100 publications), Inverse Difference Moment or Local Homogeneity $f_5$ (16,000 publications), and Entropy $f_9$ (19,100 publications) (*Haralick, Shanmugam & Dinstein, 1973*). The remaining Haralick features are used significantly less often because they depend on marginal probabilities derived from the GLCM and require extra computational steps. For instance, SA $f_6$ (3,080 publications), the SV $f_7$ (2,720 publications), and the difference variance $f_{10}$ (2,600 publications) are cited at about one order of magnitude lower than the previous category. Consequently, their meanings are more complex to grasp. We have included entropy in this study for two reasons: to demonstrate how a logarithmic moment of GLCM is estimated and, more importantly, to illustrate that the derived marginal probabilities used for evaluating SA and sum difference can immediately apply to calculating sum entropy and difference entropy features. By advancing the theoretical understanding of these features, this work aims to enhance the applicability of Haralick features in machine learning and AI-driven texture analysis.

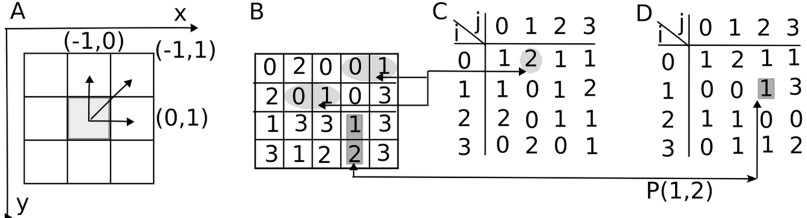

**Figure 1 Gray Level Co-occurrence Matrix (GLCM) displacement vectors.** (A) By convention, the $x$-direction runs horizontally to the right and $y$-direction vertically downward with the image's origin at the upper left corner. Pixel offsets are given by the displacement vector $d = (\Delta x, \Delta y)$. (B) In a non-periodic $N_x(= 5) \times N_y(= 4)$ 2-bit image, there are $R_x = (N_x - 1)N_y = 16$ horizontal pairs of pixels at a displacement $d = (\Delta x = 0, \Delta y = 1)$ and $R_y = N_x(N_y - 1) = 15$ vertical pairs of pixels at a displacement $d = (\Delta x = 1, \Delta y = 0)$. (C) The GLCM for unit horizontal displacement has $N_g \times N_g = 16$ non-zero entries for a 2-bit depth image. For example, the two horizontal pairs 0-1 highlighted with elliptic shades in panel B give the GLCM entry $P(0,1) = 2$. (D) The GLCM for unit vertical displacement has 15 non-zero entries. For example, the vertical pairs 1-2 indicated with rectangular shades in panel B yield the GLCM entry $P(1,2) = 1$.

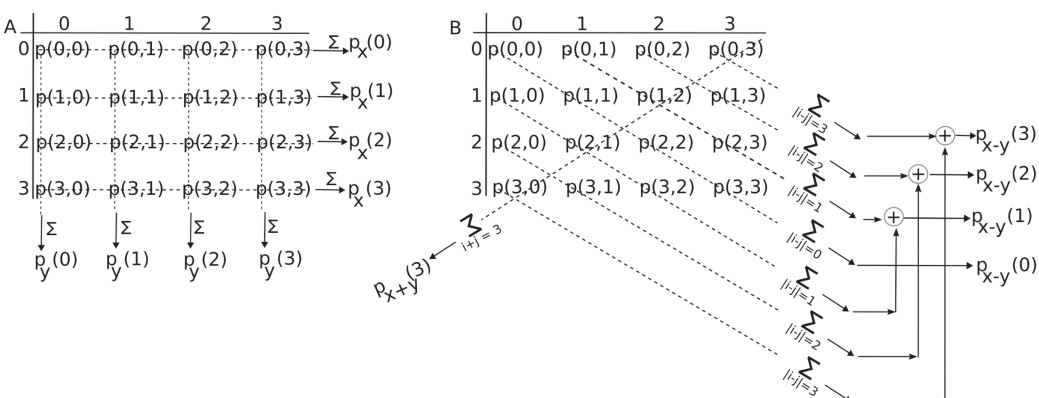

**Figure 2 Marginal probability distributions from the GLCM.** (A) The probability of finding a gray level intensity $i$ along the horizontal $x$-direction in the image is $p_x(i)$ and along vertical direction is $p_y(i)$. (B) The probability of finding a gray level difference of $k = |i - j|$ units is $p_{x-y}(k)$. It is determined by summing elements parallel to the primary diagonal at a distance of $k$ units above and below the GLCM along the corresponding dashed lines. By summing GLCM elements parallel to its secondary diagonal, one obtains $p_{x+y}(s)$.

The manuscript is structured as follows. The Methods "The Gray Level Co-occurrence Matrix (GLCM)" defines the meaning and notation for the GLCM. Figure 1 shows a reference frame attached to the upper left corner of the image and the offset vector $d = (\Delta x, \Delta y)$ between the reference (shaded) pixel and its set of neighbours. Descriptions of the x-direction $p_x(i)$, y-direction $p_y(j)$, sum $p_{x+y}(k)$, and difference $p_{x-y}(k)$ marginal distributions are provided in "Marginal Distributions Associated with the GLCM". A visual aid is included to help elucidate the meaning of the marginal distributions in Fig. 2. The numerical procedure used for generating synthetic images is detailed in "Synthetic Gradient Images". The Results section begins with a two-dimensional $N_x \times N_y$ gradient map for a periodic vertical gradient of length $N_y$ in "Two-dimensional (2D) Gradient

Maps", supporting the transition to the $N_g \times N_g$ GLCM matrix by wrapping around the 2D map in "Wrap Around the 2D Gradient Map to get the GLCM". Utilizing GLCM symmetry for periodic linear gradients enables us to estimate the number of nonzero GLCM entries for a given gradient $\nabla$ in "On the Number of Nonzero GLCM Entries for a Linear Gradient", which is necessary for calculating the marginal distribution of gray level differences $p_{x-y}(k)$ (Marginal distribution of gray level differences $p_{x-y}$ for linear gradients), the marginal distribution of gray level sums $p_{x+y}(k)$ (Marginal distribution of gray level sums $p_{x+y}$ for linear gradients). The numerical procedure used for comparing analytic predictions against numerically computed Haralick features for synthetic one-dimensional gradients is detailed in "Analytic Scaling Laws for Haralick Features of Linear Gradients. Comparison with Numerical Results". The subsequent subsections of the Results section apply the findings to derive analytic expressions and scaling laws for sum average, sum variance, difference average, and entropy dependence on $N_g$, $\nabla$ and $|d|$. Side-by-side comparison of analytical and numerical findings are summarized in the Discussion and Conclusions section.

## METHODS

### The Gray Level Co-occurrence Matrix

A grayscale image is a two-dimensional matrix $I(x, y)$ that stores gray-level intensities (see Fig. 1A). The bit depth of an image determines the number $N_g$ of gray levels. For instance, an 8-bit image has $N_g = 2^8 = 256$ gray levels. By convention, a gray level of zero, $I(x, y) = 0$, represents black, while $I(x, y) = N_g - 1$ corresponds to white. Intermediate intensities represent various shades of gray. Figure 1A shows the upper left corner reference frame attached to an image with x-direction pointing horizontally to the right and the y-direction vertically downward. Each square in Fig. 1A represents an image pixel. Arrows from the central highlighted pixel indicate the offset vectors $d = (\Delta x, \Delta y)$ to its neighbors. The increment $\Delta x$ represents the image row offset and $\Delta y$ represents the image column offset.

Figure 1B illustrates a rectangular $N_x(= 5) \times N_y(= 4)$ image with a 2-bit depth ($N_g \in \{0, 1, 2, 3\}$). For the same image, as shown in Fig. 1B, each displacement vector $d$ defines a corresponding GLCM. For instance, a unit displacement along horizontal direction $d = (\Delta x = 1, \Delta y = 0)$ produces Fig. 1C. Indeed, there are two pairs of pixels with the starting point gray level $i = 0$ and endpoint intensity level $j = 1$ separated by one pixel displacement along the horizontal direction. The array coordinates (1,4)-(1,5) and (2,2)-(2,3) are marked with elliptical shaded area and connected by the two horizontal lines extending from panel B image to the corresponding GLCM entry $P(0, 1) = 2$ in panel C. Similarly, there is only one pair of pixels in the Fig. 1B image with the starting point gray level $i = 1$ and endpoint intensity level $j = 2$ separated by one pixel displacement along the vertical direction. The array coordinates (3,4)-(4,4) are marked with rectangular shaded area and connected horizontally by a line extending from panel B image to the corresponding GLCM entry $P(1, 2) = 1$ in panel D. The unnormalized GLCM counts the number of occurrences of the (reference) gray level $i$ at a distance specified by the

displacement vector $d = (\Delta x, \Delta y)$ from the (target) gray level $j$ (*Haralick, Shanmugam & Dinstein, 1973*):

$$P_d(i,j) = \#\{((x_i, y_i), (x_j, y_j)) : I(x_i, y_i) = i \& I(x_j, y_j) = j\}, \tag{1}$$

where # denotes the number of elements in the set, the coordinates of the reference gray level $i$ are $(x_i, y_i)$, and the coordinates of the neighbor (target) pixel with gray level $j$ are $(x_j = x_i + \Delta x, y_j = y_i + \Delta y)$.

In the GLCM Eq. (1), the first index $i$ represents the intensity of the reference point, or the starting point of the displacement vector $d$, while the second index $j$ corresponds to the intensity of the endpoint of the displacement vector. For instance, an offset $d = (\Delta x = 1, \Delta y = 0)$ indicates that the row index (in the $y$-direction) remains unchanged since $\Delta y = 0$, and the column index (in the $x$- or horizontal direction across the image) increases by one unit ($\Delta x = 1$).

For simplicity, Fig. 1 only counts the pairs for gray levels one pixel apart along the horizontal (Fig. 1C) and vertical (Fig. 1D) directions, respectively. For example, only one pair of gray level intensities 1-0 is counted between the spatial coordinates (2,3) and (2,4) in Fig. 1B, which is shown as $P(1, 0) = 1$ in Fig. 1C GLCM. As a result, the Fig. 1B GLCM is not symmetric. In the original definition of the GLCM provided by Haralick (*Haralick, Shanmugam & Dinstein, 1973*), symmetry allows both $P(1, 2)$ and $P(2, 1)$ pairings to be counted as instances where the pixel value 1 is separated by the distance vector $d$ from the pixel value 2. Mathematically this is achieved by adding to the GLCMs in Figs. 1C and 1D their corresponding transposed arrays. In line with Haralick's definition, our implementation and all the results presented in this study used a symmetric GLCM matrix definition.

The number of possible pairs in the image typically normalizes the GLCM. For instance, in an $N_x \times N_y$ image, there are $R_x = (N_x - 1)N_y$ horizontal pairs and $R_y = (N_y - 1)N_x$ vertical pairs. In the example depicted in Fig. 1, since the image has $4 \times 5$ pixels, the GLCM normalization factors are $R_x = 16$ and $R_y = 15$. The corresponding normalized GLCM values in Fig. 1C are, for example, $p_d(0, 2) = P(0, 2)/R_x = 2/16$, and for Fig. 1D, they are $p_d(1, 2) = P_d(1, 2)/R_y = 1/15$. The unnormalized GLCM is indicated with capital letters such as $P_d(i, j)$, while its normalized version is denoted as $p_d(i, j)$:

$$p_d(i,j) = \frac{P_d(i,j)}{\sum_{i=0}^{N_g-1} \sum_{j=0}^{N_g-1} P_d(i,j)}. \tag{2}$$

The normalized GLCM indicates the likelihood of finding gray level $j$ at a displacement $d = (\Delta x, \Delta y)$ from the current location of the reference pixel with gray level $i$ in an image. It adheres to the normalization condition $\sum_{i=0}^{N_g-1} \sum_{j=0}^{N_g-1} p_d(i,j) = 1$. More than half of the original 14 Haralick features rely on an additional step that involves computing marginal probability distributions from $p_d(i, j)$.

The GLCM is a natural measure of image gradients, quantifying the change in light intensity from the reference intensity $i$ to the target intensity $j$ along the displacement vector $d = (\Delta x, \Delta y)$. Since Haralick features are scalar measures defined by the two-point

histogram represented by the GLCM, they also inherently measure light intensity gradients present in images.

## Marginal distributions associated with the GLCM

Only three of the original Haralick features (*Haralick, Shanmugam & Dinstein, 1973*; *Haralick, 1979*) use the normalized GLCM $p_d(i,j)$ as defined in Eq. (2). All the other use one of the four marginal probability distributions derived from $p_d(i,j)$. To simplify the notation, one dropped the subscript $d$ from the normalized GLCM $p_d(i,j)$. The $x$-direction marginal probability distribution can be obtained by summing along the rows of the GLCM $p(i,j)$:

$$p_x(i) = \sum_{j=0}^{N_g-1} p(i,j),$$

as shown in Fig. 2A. For example, $p_x(0)$ is the sum of all row elements with an intensity $i = 0$ at the reference point (see Fig. 1A), regardless of the intensity of its endpoint determined by the displacement vector. Therefore, $p_x(i)$ gives the probability of finding gray level $i$ in the image. The mean and variance of the GLCM along the marginal distribution $p_x(i)$ are $\mu_x = \sum_{i=0}^{N_g-1} i p_x(i)$, and $\sigma_x^2 = \sum_{i=0}^{N_g-1} (i - \mu_x)^2 p_x(i)$.

The $y$-direction marginal probability distribution $p_y(i)$ can be obtained by summing the columns of the GLCM $p(i,j)$:

$$p_y(j) = \sum_{i=0}^{N_g-1} p(i,j).$$

For example, $p_y(0)$ is the sum of all column elements with an endpoint intensity $j = 0$, regardless of the intensity of the reference (starting) point. These marginal probabilities are illustrated in Fig. 2, along the horizontal dashed lines representing the GLCM line summation for $p_x$ and along the vertical dashed lines representing the GLCM column summation of $p(i,j)$ to obtain $p_y$, respectively.

The marginal distribution of gray level differences $k = i - j$ between the reference pixel intensity $i$ and the endpoint intensity $j$ determined by the displacement vector $d$ is:

$$p_{x-y}(k) = \sum_{i=0}^{N_g-1} \sum_{j=0}^{N_g-1} \delta_{|i-j|,k} p(i,j), \tag{3}$$

where $\delta_{m,n}$ is Kronecker's symbol. For example, $p_{x-y}(0)$ represents the sum of all primary diagonal elements of the GLCM, as these elements exhibit no gray level differences between the reference point and the endpoint of the vector $d$, as illustrated in Fig. 2B. Similarly, the sum of the elements along the first line parallel to and above the primary diagonal reflects a gray level difference of $k = +1$ units between the reference gray level $i$ and the endpoint gray level $j$ of the GLCM, which defines $p_{x-y}(1)$. The sum $p(0,1) + p(1,2) + p(2,3)$ of GLCM entries along the first line parallel and above the primary diagonal in Fig. 2B correspond to the fraction of $p_{x-y}(1)$ with $j - i = +1$. The

sum $p(1,0) + p(2,1) + p(3,2)$ of GLCM entries along the first line parallel and below the primary diagonal in Fig. 2B correspond to the fraction of $p_{x-y}(1)$ with $j - i = -1$. Since the definition of gray level differences marginal distribution $p_{x-y}(1)$ in Eq. (3) counts absolute differences $k = |i - j|$, the two partial sums must also be added (see the $\oplus$ symbol) to produce $p_{x-y}(1)$.

The marginal distribution of gray level sums $k = i + j$ between the reference pixel intensity $i$ and the endpoint neighbor intensity $j$ is:

$$p_{x+y}(k) = \sum_{i=0}^{N_g-1} \sum_{j=0}^{N_g-1} \delta_{i+j,k} p(i,j). \tag{4}$$

To prevent overcrowding in Fig. 2B, we only showed the $p_{x+y}(3)$, which signifies the sum of the GLCM elements along its secondary diagonal with $i + j = 3$, *i.e.*, $p(3,0) + (2,1) + p(1,2) + p(0,3)$. Other values for $p_{x+y}(s)$ correspond to summation along lines parallel to the secondary diagonal in Fig. 2B.

## Synthetic gradient images

While the GLCM method described in "Methods" applies to any image, this study specifically focuses on computer-generated (synthetic) images with one-dimensional vertical gradients. This focus is motivated by the fact that image gradients are highly invariant across images (*Long & Purves, 2003*; *Tward, 2021*; *Dresp-Langley & Reeves, 2024*).

Image gradients have long been used as statistical (or spectral) priors for estimating image features (*Gong & Sbalzarini, 2014, 2016*). A gradient image $G(x, y)$, is derived from the first-order spatial differences of the original image, $I(x, y)$, such that $G(x, y) = (I(x - 1, y) - I(x, y), I(x, y - 1) - I(x, y))$ (*McCann & Pollard, 2008*; *Sevcenco & Agathoklis, 2021*). The gradient image retains the same dimensions as the original but stores the x- and y-direction gradient values at each pixel.

Gradient spectral priors have been extensively applied in various image processing tasks, including denoising and deblurring (*Chen, Yang & Wu, 2010*), image restoration (*Cho et al., 2012*), range compression (*Fattal, Lischinski & Werman, 2002*), shadow removal (*Finlayson, Hordley & Drew, 2002*), and image compositing (*Levin et al., 2004*; *Perez, Gangnet & Blake, 2003*). Notably, deblurring in the gradient domain is often more computationally efficient than operating on raw pixel values (*Cho & Lee, 2009*; *Shan, Jia & Agarwala, 2008*; *Wang & Cheng, 2016*).

Traditionally, images are decomposed into 2D orthogonal gradient maps assuming that x- and y-direction gradients are statistically independent. One of the first studies to explore potential correlations between these gradient distributions in natural scene images found "weakly negatively correlated in the training dataset (from edges in the images)" (*Gong & Sbalzarini, 2016*). Consistent with these findings, recent algorithms for image denoising and deblurring (*Zheng et al., 2022*; *Zhangying et al., 2024*), range compression (*Yan, Sun & Davis, 2024*), or pattern classification (*Wang et al., 2025*) continue to treat orthogonal gradients as independent and their spectral priors as uncorrelated. Based on

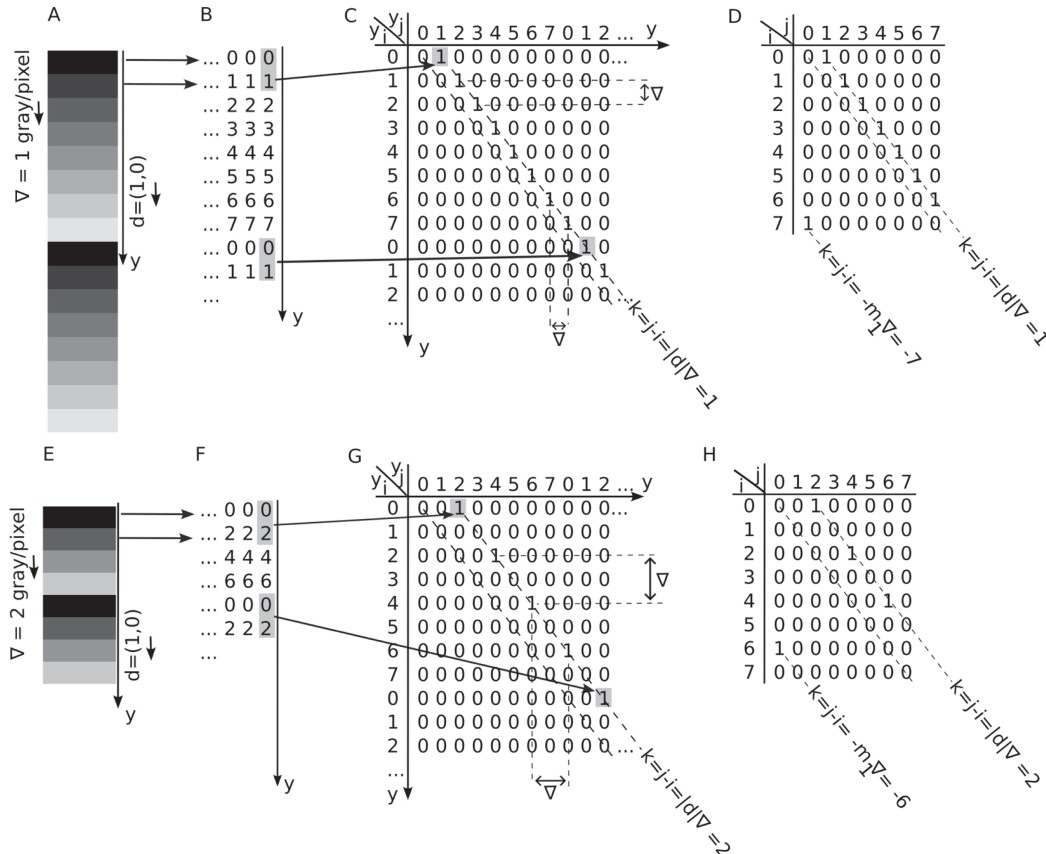

**Figure 3 Periodic and linear vertical gradients and their GLCM.** (A and E) Horizontal stripes of constant intensity with a periodic vertical gradient of $\nabla = 1$ (panel A) and $\nabla = 2$ (panel 2) gray levels per pixel in a $b = 3$-bit depth grayscale image. Each horizontal line is one pixel wide. (B and F) Numerical representation of the grayscale image with values ranging from zero to $N_g - 1$. (C and G) The two-dimensional (2D) gradient map of the periodic gray level gradient displays nonzero entries at the coordinates $(y_i, y_j) = (y_i, y_{i+|d|})$, which are spaced by the distance $d$ and maintain the absolute coordinates of pixels along the gradient. The first non-zero entry occurs at $(i = 0, j = d)\nabla)$, with all nonzero entries separated by distances of $\nabla$ both vertically and horizontally. (D and H) The shaded gray levels $i = 0$ and $j = 1$ at a vertical distance of one pixel $d = (0, 1)$ in panel B determine the GLCM entry $P(0, 1) = 1$. The GLCM can be obtained by wrapping around the 2D gradient map by modulo $N_g + 1$ in both array dimensions.

this well-supported assumption, our study focuses exclusively on a vertical gradient for calculating Haralick texture features.

Figure 3A shows a $b = 3$-bit depth grayscale image with dimensions $N_x \times N_y$, featuring a vertical, linearly increasing, periodic intensity gradient of $\nabla = 1$ gray level per pixel. The array $I(x, y)$ that represents the image is given by $I(x_i, y_i) = y_i \nabla$ where $y_i = \{0, 2, ..., N_y - 1\}$. Since image intensities do not depend on the $x_i = \{0, 2, ..., N_x - 1\}$ matrix index, the image appears as horizontal stripes with linearly increasing intensity (Fig. 3A). Furthermore, the vertical gradients are periodic, *i.e.*, the intensity pattern repeats after reaching the maximum number of gray levels $N_g = 2^b$. In other words, the vertical coordinate $y_i$ and pixel intensity are connected through

$I(x_i, y_i) = mod(y_i, N_g)\nabla$. The modulo ("mod") operation along the vertical spatial indices $y_i$ ensures the gradient repeats periodically after $N_g$ pixels. In Fig. 3A example, the gray levels increase linearly from zero to $N_g - 1$ with a step of $\nabla = 1$ gray level per pixel. The arrow next to the gradient in Fig. 3A indicates the gradient's direction. Similarly, Fig. 3E shows a synthetically generated image with a vertical, linearly increasing, and periodic gradient $\nabla = 2$ gray levels per pixel. The grayscale images from Fig. 3A and Fig. 3E are numerically represented in Fig. 3B and Fig. 3F, respectively. The horizontal arrows between panels A and B indicate that the constant intensity line of pixels is represented numerically by the corresponding integer values with black mapped to 0. Following the procedure described above, we generated square synthetic images of $1024 \times 1024$ pixels containing periodic linear gradient patterns, as illustrated in Fig. 3. Our analysis focuses on three key variables:

(1) The number of gray levels in the image ($N_g$),
(2) The intensity of the image gradients ($\nabla$) in gray levels per pixel, and
(3) The displacement vector ($d = (\Delta x, \Delta y)$) in pixels, which determines the GLCM matrix used to compute the Haralick features.

We created images with a bit depth (b) ranging from 4 to 8, corresponding to $N_g = 2^b \in \{16, 32, 64, 128, 256\}$. These values represent a broad and realistic range for evaluating how Haralick features depend on $N_g$ (see Figs. 4 and 5). For each bit depth we generated synthetic images with gradient intensities ($\nabla$) ranging from 1 to 8. However, to reduce visual clutter, only odd $\nabla$ values are displayed in Figs. 4 and 5. Finally, for each combination of bit depth ($b$) and gradient intensity $\nabla$, we computed GLCMs for vertical displacement vectors $|d| = 1, \ldots, 8$.

## RESULTS

Interpreting GLCM and Haralick features is difficult because they contain second-order statistical information about image pixels. To calculate Haralick features, one employs images with a single periodic and linear gradient to understand the relationship between image gradients and GLCM symmetries.

### Two-dimensional gradient maps

To count the pairs of pixels with a starting gray level $i$ and an endpoint gray level $j$ separated by a distance $d = (\Delta x, \Delta y)$ pixels, one can create a two-dimensional (2D) $N_y \times N_y$ gradient map such that its $(y_i, y_j) = (y_i, y_{j=i+|d|})$ entry is 1 if $I(x_i, y_{j=i+|d|}) - I(x_i, y_i) = d \cdot \nabla$ and zero otherwise as shown in Fig. 3C. Here, $\cdot$ is the dot product and ensures that one considers the relative orientation of the gradient $\nabla$ to the displacement vector $d$. From Figs. 3B and 3F, one notices that the gray level intensity at spatial coordinate $y_i$ is always $y_i = i\nabla$ with $i = 0, \ldots, N_y - 1$. The pixel intensity at a vertical coordinate $y_j$, which is a distance $|d|$ from $y_i$, is $y_j = y_i + d \cdot \nabla = (i + |d|)\nabla$. As a result, the 2D maps in Figs. 3C and 3G are one-to-one correspondences between pixel location $y_i$ and its corresponding gray level intensity $i\nabla$. One notices in Fig. 3C with $\nabla = 1$

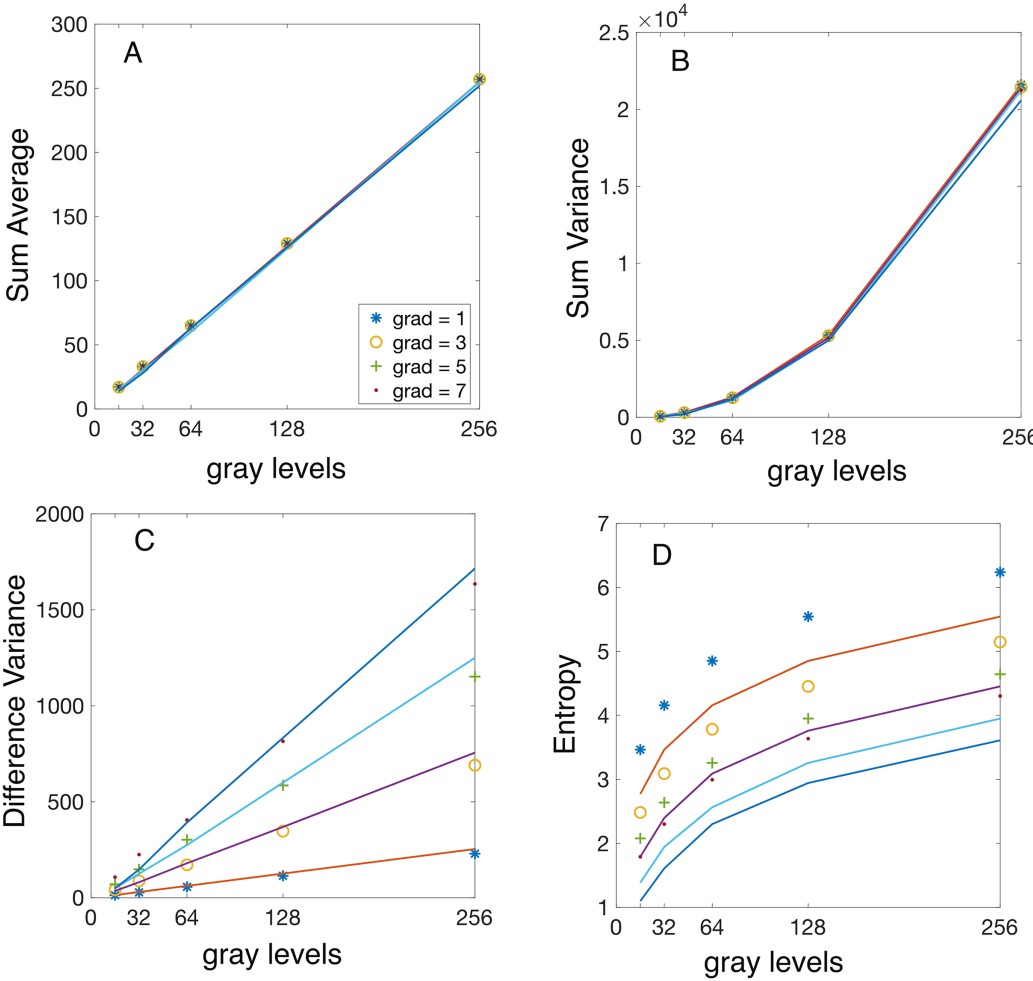

**Figure 4 Analytical *vs* numerically calculated features scaling with image bit depth.** Synthetic linear gradient images were used with $N_g \in \{16, 32, 64, 128, 256\}$ gray levels. The GLCMs were numerically evaluated for a fixed integer vertical displacement $|d| = 1$ pixels and variable linear gradients of $\nabla = 1$ gray level per pixel (symbol "∗"), $\nabla = 3$ gray levels per pixel (symbol "o"), $\nabla = 5$ gray levels per pixel (symbol "+"), and $\nabla = 7$ gray levels per pixel (symbol "."). All Haralick features were computed numerically using Matlab's *graycoprops()* function. The continuous lines represent the analytically predicted scaling laws for the corresponding features. (A) The numerically computed sum average (SA) feature $f_6$ increases linearly with $N_g$ and is independent of the magnitude of the displacement vector and the gradient. (B) The numerically computed sum variance (SV) feature $f_7$ exhibits a quadratic dependence on the magnitude of the displacement vector and is independent of the magnitude of the displacement vector and the gradient, as predicted by Eq. (13). (C) The numerically computed difference variance (DV) $f_{10}$ scales linearly with the image bit depth and the slope increases linearly with the image gradient intensity $\nabla$, as predicted by Eq. (15). (D) The experimental values of entropy $f_9$ show the predicted logarithmic trend, but they are consistently and slightly shifted in comparison to the theoretical prediction from Eq. (20). The reason is that the numerically computed Entropy feature uses $\log(p(i, i) + \varepsilon)$ with a small $\varepsilon$ constant to prevent logarithm divergence for sparse GLCM with many $p(i, i) = 0$.

gray level per pixel and Fig. 3G with $\nabla = 2$ gray level per pixel that the vertical and horizontal distance between any non-zero entries of the 2D gradient map is $\nabla$. These $\nabla$ displacements are marked in Fig. 3C and Fig. 3G, respectively. Additionally, one can

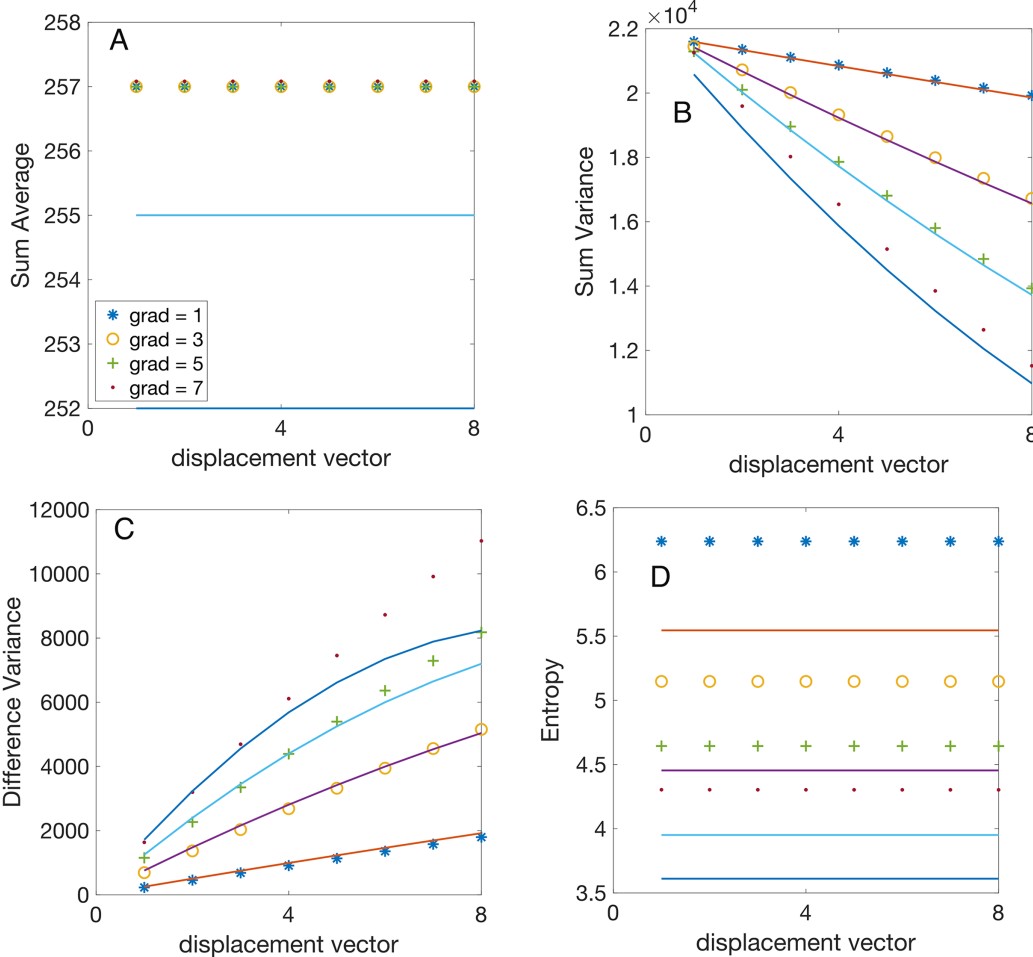

**Figure 5 Analytical *vs* numerically calculated features scaling with displacement vector magnitude.** All synthetic gradient images were 8-bit depth. The GLCMs were numerically evaluated for vertical displacements $|d| = 1, \ldots, 8$ pixels and linear gradients $\nabla = 1$ gray level per pixel (symbol "*"), $\nabla = 3$ gray levels per pixel (symbol "o"), $\nabla = 5$ gray levels per pixel (symbol "+"), and $\nabla = 7$ gray levels per pixel (symbol "."). All features were numerically computed using Matlab's function *graycoprops()*. The continuous lines illustrate the analytically predicted scaling laws for the corresponding features. (A) The numerically computed sum average (SA) feature $f_6$ remains independent of the magnitude of the displacement vector and exhibits negligible gradient dependence due to the integer part function, as elaborated in the text. (B) The numerically computed sum variance (SV) feature $f_7$ scales linearly with the magnitude of the displacement vector, with a slope proportional to the gradient, as predicted by Eq. (13). (C) The numerically computed difference variance (DV) $f_{10}$ scales linearly with the magnitude of the displacement vector and the slope is proportional to the gradient, as predicted by Eq. (16). (D) The experimental values of entropy $f_9$ are independent of the magnitude of the displacement vector and increase with the gradient, as expected from Eq. (20). The slight systematic difference between the computed and predicted values is due to the actual entropy feature calculation using $\log(p(i, i) + \varepsilon)$ with a small $\varepsilon$ constant to prevent logarithm divergence for sparse GLCM with many $p(i, i) = 0$.

observe from the 2D gradient maps in Figs. 3C and 3G that all nonzero entries are aligned with the primary diagonal of the 2D gradient map at a distance of $d \cdot \nabla$ from it. The distance of the gradient pattern from the primary diagonal of the 2D gradient maps is determined by the first gray level intensity, *i.e.*, $i = 0$, which is always paired with the gray

label $j = d \cdot \nabla$ for any displacement vector $d$ and gradient intensity $\nabla$. Finally, all nonzero entries $(y_i, y_j)$ in the 2D gradient maps shown in Figs. 3C and 3G obey the condition $k = |i - j| = d \cdot \nabla$ shown with dashed line parallel to the principal axis diagonal. The principal diagonal elements are always zero because they correspond to a uniform image with no intensity changes from pixel to pixel.

## Wrap around the 2D gradient map to get the GLCM

While illuminating, representing a periodic linear gradient of length $N_y$ using a sparse $N_y \times N_y$ 2D gradient maps, as shown in Figs. 3C and 3G, is not efficient. As a result, the GLCM removes the extra spatial information about pixel coordinates $(y_i, y_j)$ retained by the 2D gradient map and only counts the co-occurrence of gray level intensities $i$ and $j$ at a relative distance $d = (\Delta x, \Delta y)$, as illustrated in Figs. 3D and 3H. Consequently, for a specific displacement vector $d$, the GLCM is an $N_g \times N_g$ matrix that solely counts the co-occurrence of gray levels $i$ and $j$ at a relative distance $d$ from each other, irrespective of their absolute spatial coordinates $y_i$ and $y_j$. Because the absolute coordinates $(y_i, y_j)$ of the pixel intensity pair $i$ and $j$ are no longer recorded, the GLCM is not a one-to-one mapping of the original gradient (unlike the 2D gradient map). For instance, in Fig. 3C, the pixel intensities $i = 7$ and $j = 0$ are located at a distance $d = (0, 1)$, and they are represented in the 2D gradient map by a value of "1" at spatial coordinates $(y_i = 7, y_j = 8)$, as shown in Figs. 3A and 3B. However, the GLCM represents the same pair as an entry at $(i = 7, j = 0)$ as it remaps all 2D gradient map entries from Figs. 3C and 3G using a modulo $N_g$ operation. For example, the spatial coordinates $(y_i = 7, y_j = 0)$ from Fig. 3C are mapped modulo $N_g + 1 = 9$ to GLCM coordinates $(y_i = 7, y_j = 0)$, which correspond to gray levels $(i = 7, j = 0)$ in GLCM. Although the $N_g \times N_g$ GLCM array can no longer be mapped back to the original image, it retains essential second-order spatial correlations of gray level intensities.

## On the number of nonzero GLCM entries for a linear gradient

For any linear gradient $\nabla$, the starting point of the GLCM has an index $i$ from the set $\{0, \nabla, 2\nabla, \ldots, (\tilde{N}_g - 1)\nabla\}$, where $\tilde{N}_g$ is the number of non-zero GLCM entries shown in Figs. 3D and 3H, *i.e.*:

$$\tilde{N}_g = 1 + \left[\frac{N_g - 1}{|\nabla|}\right]. \tag{5}$$

In the above formula, $[\ldots]$ denotes the integer part. Each endpoint index $j$ of the GLCM is also expressed as $j = i + \nabla$. This relationship indicates that the increment of endpoint indices in the GLCM, represented by $\Delta j$, is equivalent to that of the starting point indices, denoted as $\Delta i$, meaning that $\Delta j = \Delta i = \nabla$, as illustrated by the horizontal and vertical double arrows in Fig. 3C for $\nabla = 1$ and in Fig. 3G for $\nabla = 2$. For example, in Fig. 3D and $\nabla = 1$ gray level per pixel along with Eq. (5) determines how many non-zero GLMC entries $\tilde{N}_g$ result from sampling the $N_g = 8$ gray levels of the image, *i.e.*, $\tilde{N}_g = 1 + [(8 - 1)/1] = 8$. Similarly, for Fig. 3H with $\nabla = 2$ gray levels per pixel in conjunction with Eq. (5), it yields $\tilde{N}_g = 1 + [(8 - 1)/2] = 4$.

One can notice that Fig. 3 displays the GLCMs for positive gradients $\nabla > 0$ and positive displacement vectors, such as $d = (\Delta x, \Delta y) = (0, 1)$. Reversing the direction of the gradient would merely shift all non-zero entries in the two-dimensional representation shown in Figs. 3C and 3G below the primary diagonal at a distance $i - j = d \cdot \nabla < 0$.

**Marginal distribution of gray level differences $p_{x-y}$ for linear gradients**
The marginal probability distribution $p_{x-y}(k)$, defined by Eq. (3) and visually represented in Fig. 2B, accounts for the sum of GLCM entries with specified gray level differences $k = j - i$. As observed in Figs. 3D and 3H, the lines parallel to the primary diagonal of the GLCM convey information about image gradients and represent the lines of constant gray level differences $p_{x-y}(k)$. For example, the GLCM primary diagonal entries have zero gray level differences, *i.e.*, $k = j - i = 0$. Consequently, the sum of the primary diagonal elements, *i.e.*, $p_{x-y}(0) = \sum_{i=0}^{N_g - 1} \sum_{j=0, |i-j|=0}^{N_g - 1} p(i, j)$, is a zero gradient line because the gray level differences, *i.e.*, the difference between the gray level value $i$ of the start (reference) point $(x_i, y_i)$ and the endpoint gray level intensity $j$ at $(y_j, x_j)$ along the displacement vector $d = (\Delta x, \Delta y)$, is $k = |i - j| = 0$. Figure 3D illustrates that the GLCM of a gradient $\nabla = 1$ gray level per pixel along the vertical unit displacement vector $d = (0, 1)$ contains all entries (except one) aligned along a parallel line with the primary diagonal at gray level differences $k = j - i = d \cdot \nabla = 1$. The sole exception is the GLCM entry at the discontinuity between the first period and the subsequent gradient repeats (see Fig. 3A and Fig. 3E). For example, the first period of the gradient in Fig. 3E and Fig. 3F ends with a gray level of $i = 6$ in an image with $N_g = 8$ gray levels and a gradient intensity $\nabla = 2$. Therefore, its pair must have an intensity $j = i + \nabla = 8$, which is mapped modulo $N_g$ to $j = 0$. It corresponds to $P(6, 0) = 1$ (remember that the wrapping around of GLCM in gray level spaces is done modulo $N_g$ because the gray level indices start at zero while the spatial coordinates wrap around with modulo $N_g + 1$ operation because they begin at index 1). Since accounting for another period of the same gradient increases all nonzero entries of GLCM by one unit, from this point forward, one only calculates the GLCM for a single period of the gradient. To compute Haralick's features, one uses the symmetry of the GLCM induced by periodic linear gradients such as those shown in Fig. 3.

One can observe from Fig. 3 that the nonzero GLCM entries parallel to the primary diagonal for a given gray level difference $k = j - i = d \cdot \nabla$ begin at a distance of $d \cdot \nabla$ from the first GLCM entry $p(0, 0)$. The line of constant gray level differences $k = j - i = d \cdot \nabla$ (dotted line parallel to the primary diagonal of the GLCM in Figs. 3D and 3H) starts at $p(0, d \cdot \nabla)$ and ends at $p(i = (m_1 - 1)\nabla, j = i + d \cdot \nabla)$, where $m_1$ is the number of GLCM entries along the gray level differences line with $k = j - i = d \cdot \nabla$, which is:

$$m_1 = \tilde{N}_g - |d|. \tag{6}$$

In the example depicted in Figs. 3A–3D, the GLCM for a unit vertical displacement $d = (0, 1)$ in an image exhibiting a linear gradient of $\nabla = 1$ gray level per pixel and $N_g = 8$ gray levels has a total of $\tilde{N}_g = 8$ nonzero entries (from Eq. (5)), of which $m_1 = 7$

(see Eq. (6)) along the line of constant gray level differences $k = j - i = d \cdot \nabla = 1$. This line starts at $p(0, d \cdot \nabla) = p(0, 1)$ and ends at $p(i = (m_1 - 1)\nabla, j = i + d \cdot \nabla) = p(6, 7)$. Similarly, for the example shown in Figs. 3E–3H, $\nabla = 2$ gray levels per pixel and $N_g = 8$, one gets a total number of GLMC entries of $\tilde{N}_g = 4$ (from Eq. (5)), of which $m_1 = 3$ (see Eq. (6)) along the line of constant gray level differences $k = j - i = d \cdot \nabla = 2$ that starts at $p(0, d \cdot \nabla) = p(0, 2)$ and end at $p(i = (m_1 - 1)\nabla, j = i + d \cdot \nabla) = p(4, 6)$.

The GLCM always has exactly $\tilde{N}_g$ nonzero entries according to Eq. (5), of which, according to Eq. (6), $m_1$ are on the constant gray level differences line $k = j - i = d \cdot \nabla$. The remaining $m_2$ nonzero GLCM entries have the endpoint coordinate $j$ always beginning at zero due to the wrapping around modulo $N_g$ in the gray level intensity space:

$$m_2 = \tilde{N}_g - m_1 = |d|. \tag{7}$$

Such GLCM entries are $p(i = m_1\nabla, j = 0)$, $p(i = (m_1 + 1)\nabla, j = \nabla)$, and so on. One notices that all these new $m_2 = |d|$ GLCM entries align along the line of constant gray level differences $k = j - i = -m_1\nabla$, as shown in Figs. 3D and 3H. To summarize, the (unnormalized) marginal distribution of gray level differences $p_{x-y}$ for linear gradients represents the frequency of various combinations of pixel intensities that yield a specific absolute difference value $k = |j - i|$:

$$p_{x-y}(k) = \begin{cases} 1, & \text{for } k = j - i = d \cdot \nabla \text{ with } i = \{0, 1, \ldots, m_1-1\}\nabla, \\ 1, & \text{for } k = j - i = -m_1\nabla \text{ with } i = \{m_1, m_1 + 1, \ldots \tilde{N}_g - m_1\}\nabla, \\ 0, & \text{otherwise.} \end{cases} \tag{8}$$

## Marginal distribution of gray level sums $p_{x+y}$ for linear gradients

The previous section demonstrated that linear gradients are naturally represented by non-zero entries parallel to the primary diagonal of the GLCM. Thus, the marginal distribution of gray level difference $p_{x-y}$ arises naturally from GLCM symmetry. Other Haralick features require calculating the marginal distribution $p_{x+y}(s)$ for a given sum of gray level intensity $s = i + j$, where $s = \{0, 1, \ldots, 2(N_g - 1)\}$. One can utilize the GLCM symmetries caused by linear gradients and the corresponding marginal distribution $p_{x-y}(k)$ where $k = j - i = d \cdot \nabla$ to streamline the calculation of the other marginal distribution $p_{x+y}$. Indeed, from $p_{x-y}(k)$, the $m_1$ nonzero endpoint gray level intensity are $j = i + k = i + d \cdot \nabla$ where $i = \{0, \nabla, \ldots, (m_1 - 1)\nabla\}$. Therefore, the elements of the marginal distribution $p_{x+y}(s)$ are $s = i + j = 2i + d \cdot \nabla$ with $i = \{0, \nabla, \ldots, (m_1 - 1)\nabla\}$. Similarly, the second line of constant gray level differences is $k = j - i = -m_1\nabla$ where $j = i - m_1\nabla$ and $i = \{m_1\nabla, (m_1 + 1)\nabla, \ldots\}$, which determines the marginal distribution $p_{x+y}(s)$ with $s = i + j = 2i - m_1\nabla$. In summary, the (un-normalized) marginal distribution of gray level sums $p_{x+y}$ for linear gradients indicates the frequency of various combinations of pixel intensities that total a specific value $s = j + i$:

$$p_{x+y}(s) = \begin{cases} 1, & \text{for } s = 2i + d \cdot \nabla \text{ with } i = \{0, 1, \ldots, m_1-1\}\nabla, \\ 1, & \text{for } s = 2i - m_1\nabla \text{ with } i = \{m_1, m_1 + 1, \ldots \tilde{N}_g - m_1\}\nabla, \\ 0, & \text{otherwise.} \end{cases} \tag{9}$$

## Analytic scaling laws for Haralick features of linear gradients
## Comparison with numerical results

The previous subsection includes all the elements needed to estimate analytically any Haralick feature. In the following subsections, we derive analytical formulas for SA, SV, difference variance (DV), and Entropy based on the GLCMs symmetries derived in the previous subsections. Anticipating the results from the following subsections, the analytic scaling laws for Haralick features take the general form

$$f \propto N_g^\alpha |d|^\beta \nabla^\gamma,$$

where the scaling exponents $\alpha$, $\beta$ and $\gamma$ are derived from the GLCM symmetries as we will prove below.

To validate our theoretically predicted scaling laws for Haralick features, we performed numerical calculations using synthetic (computer-generated) gradient images. The predictions are represented by continuous lines in Figs. 4 and 5. At the same time, the corresponding numerical simulation results—based on the synthetic images described in "Synthetic Gradient Images"—are shown as discrete points with different symbols, as indicated in the figure legends.

To reduce plot clutter in Figs. 4 and 5, we present results only for odd intensity gradient values of $\nabla \in \{1, 3, 5, 7\}$ gray levels per pixels. In Fig. 4 the displacement vector magnitude was fixed at $|d| = 1$, while the number of gray levels varied as $N_g \in \{16, 32, 64, 128, 256\}$. Conversely, in Fig. 5 the bit depth was set to $b = 8$ bits ($N_g = 256$), while the vertical displacement vector magnitude varied as $|d| = 1, \dots, 8$. For each synthetic image with a given bit depth $b$ and gradient intensity $\nabla$, we computed the GLCMs for each vertical displacement vector $d$ using Matlab's *graycomatrix*() function. For instance, the GLCM shown in Fig. 1C was obtained using *graycomatrix*(img,'Offset',[0 1], 'NumLevels', 4, 'GrayLimits', [], 'Symmetric',false). Additionally, when calculating all Haralick features, we consistently set the 'Symmetric' flag in graycomatrix() to true. Subsequently, we computed Haralick features from GLCMs using Matlab function *graycoprops*().

For a single period of a linear gradient $\nabla$ (see Fig. 3) all $\tilde{N}_g$ nonzero entries of the GLCM given by Eq. (5) have equal weight and are only aligned to two parallel lines to the primary diagonal as in Figs. 3D and 3H.

### Sum average $f_6$

The SA indicates the uniformity of intensity values across the image texture. A higher SA value represents an even distribution of intensity sums between neighboring pixels. SA is defined as:

$$f_6 = \sum_{k=0}^{2(N_g-1)} k p_{x+y}(k).$$

(10)

A high SA implies that most pixel pairs have similar intensity sums, indicating a relatively uniform texture. A low SA suggests more significant variation in intensity

sums between neighboring pixels, signifying a more textured appearance. From Eq. (10) with Eq. (9)

$$f_6 = \frac{1}{\tilde{N}_g}\left(\overbrace{d\nabla + (d\nabla + 2\nabla) + \dots (d\nabla + 2(m_1-1)\nabla)+}^{k=j-i=d\nabla}\right.$$

$$\left.\underbrace{m_1\nabla + (m_1\nabla + 2\nabla) + \dots (m_1\nabla + 2(\tilde{N}_g - m_1)\nabla)}_{k=j-i=-m_1\nabla}\right) = (\tilde{N}_g - 1)\nabla = \nabla\left[\frac{N_g - 1}{\nabla}\right]. \quad (11)$$

To simplify the calculation of $f_6$ above, we separated the contributions of the GLCM entries that are parallel to its primary diagonal at a distance of $k = j - i = d\nabla$ from those on the line where $k = j - i = -m_1\nabla$. Each of the two terms in Eq. (11) is an arithmetic series with the sum $\sum_{q=0}^{Q} a + 2\nabla q = a(Q+1) + Q(Q+1)\nabla$. For $k = j - i = d\nabla$ in Eq. (11) one uses $a = d\nabla$ and $Q = m_1 - 1$ for $k = j - i = -m_1\nabla$ one substitute $a = m_1\nabla$ and $Q = \tilde{N}_g - m_1$.

The first observation is that the theoretically predicted SA value given by Eq. (11) is independent of the gradient intensity $\nabla$ (see the continuous lines in Fig. 4A) and the displacement vector $d$ (see the continuous lines in Fig. 5A) as summarized also in Table 1. Numerically computed Haralick feature SA confirms that its values are independent of gray level intensity gradients $\nabla$ and increases linearly with the number of gray levels $N_g$ as shown in Fig. 4A. The exact formula in Eq. (11), which involves the discontinuous integer part function $[\dots]$, is challenging to work with; however, by dropping the integer part operation, one finds a continuous approximate value $\tilde{f}_6 \approx N_g - 1$. This approximation demonstrates that $f_6$ scales linearly with $N_g$ (see the continuous lines Fig. 4A, which is also confirmed numerically by the linear increase of Haralick features with the number $N_g$ of gray levels shown in Fig. 4A. The second observation is that, numerical simulations shown in Fig. 5A confirm our theoretical prediction based on Eq. (11) that SA feature is independent of the displacement vector magnitude $|d|$. One notices, a slight error in approximating $f_6$ with $\tilde{f}_6$. For example, a $N_g = 256$ gray level image and a gradient $\nabla = 2$ gray levels per pixel gives $f_6 = 2[(256 - 1)/2] = 254$, which is slightly less than the simplified approximation $\tilde{f}_6 = 255$, but the error is under 0.4%. Even for gradients as large as $\nabla = 10$ gray levels per pixel, the error of approximating $f_6$ with $\tilde{f}_6 \approx N_g - 1$ is below 1%. This slight disagreement between the theoretical predicted SA value from Eq. (11) and the numerically computed values is emphasized in Fig. 5A. One can conclude that the gradient $\nabla$ slightly decreases the SA value $f_6$, but the correction is negligible for small gradients $\nabla < 10$ gray levels per pixel. This fact is marked by the general attribute "independent" with an asterisk in Table 1.

## Sum variance

The sum variance feature is defined as follows:

$$f_7 = \sum_{k=0}^{2(N_g-1)} (k - f_6)^2 p_{x+y}(k), \quad (12)$$

**Table 1 Summary of feature scaling laws $f \propto N_g^\alpha |d|^\beta \nabla^\gamma$.**

|  | $N_g$ | $|d|$ | $\nabla$ |
|---|---|---|---|
| Sum average | Linear | Independent | Independent* |
| Sum variance | Quadratic | Linear | Linear |
| Difference variance | Linear | Linear | Linear |
| Entropy | Logarithmic | Independent | Independent* |

**Note:**
The asterisk mark next to "independent" attribute means the respective feature very slightly decreases with $\nabla$, and this effect can be neglected for $\nabla < 10$ gray levels per pixel.

and can be analytically estimated for GLCM of linear gradients using the same strategy described above when deriving explicit analytical expression for SA in Eq. (11).

$$f_7 = \frac{1}{\tilde{N}_g} \left( \overbrace{(d\nabla - f_6)^2 + (d\nabla + 2\nabla - f_6)^2 + \ldots (d\nabla + 2(m_1 - 1)\nabla - f_6)^2}^{k=j-i=d\nabla} + \right.$$

$$\left. \underbrace{(m_1\nabla - f_6)^2 + (m_1\nabla + 2\nabla - f_6)^2 + \ldots (m_1\nabla + 2(\tilde{N}_g - m_1)\nabla - f_6)^2}_{k=j-i=-m_1\nabla} \right) =$$

$$\nabla^2(\tilde{N}_g^2/3 - \tilde{N}_g d + d^2 - 1/3). \tag{13}$$

To accurately predict the scaling law of SV features from the exact formula given by Eq. (13), one could eliminate the integer part function from the definition of $\tilde{N}_g$ and utilize an approximate estimate:

$$\tilde{f}_7 = ((N_g - 1)^2/3 - (N_g - 1)d\nabla + d^2\nabla^2 - 1/3\nabla^2). \tag{14}$$

The discrepancy between the true $f_7$ (Eq. (13)) and the approximate estimate $\tilde{f}_7$ is minor but can reach several percentage points. For example, the largerst error occurs for $N_g = 256$, $\nabla = 7$, and $|d| = 8$, which is approximately 3.33%.

Based on Eq. (14), one notice that SV scales quadratically with $N_g$. Indeed, the second term in Eq. (14), which is linear in $N_g$, is always smaller than the first term, quadratic in $N_g$ if $d\nabla < N_g$. This condition is fulfilled because the product $d$ pixels times $\nabla$ gray levels per pixel is the number of gray levels variation across an image, which cannot be larger than $N_g$. Numerical simulations confirmed our analytical prediction of a quadratic scaling law for $f_7$ with $N_g$, as shown in Fig. 4B. One also notices from Fig. 4B that for fixed displacement vector magnitude $|d|$, numerical values of SV are independent of gradient intensity $\nabla$ as predicted analytically by Eq. (14).

For an image with a fixed number of gray levels $N_g$ and gradient $\nabla$ gray levels per pixel, the second term in Eq. (14) dominated SV's dependence on $|d|$. This is because $(N_g - 1)d\nabla > d^2\nabla^2$, which reduces to $N_g - 1 > d\nabla$. This was shown above to be true for all images. Furthermore, the second term in Eq. (14) $(N_g - 1)d\nabla$ is also larger than the fourth term $1/3\nabla^2$ because $(N_g - 1)d > 1/3\nabla$ even for the smallest possible displacement vector with $|d| = 1$. As a result, the linear term $|d|$ is the primary influence in the scaling law of $f_7$, which aligns with our numerical simulations shown in Fig. 5B. As noticed from Fig. 5B, for

a fixed displacement vector magnitude $|d|$, SV linearly changes with the gradient intensity $\nabla$ as predicted analytically (see also Table 1).

## Difference variance

The definition of difference variance is:

$$f_{10} = \sum_{k=0}^{2(N_g-1)} (k - DA)^2 p_{x-y}(k),\tag{15}$$

where the DA is given by $DA = \sum_{k=0}^{2(N_g-1)} k p_{x-y}(k)$. The evaluation of DA is straightforward and follows from Eq. (8) since all GLCM entries are equal weight:

$$DA = \frac{1}{\tilde{N}_g}(m_1 d\nabla + m_2(-m_1\nabla)) = 0.$$

As a result, the DA reduces to

$$f_{10} = \sum_{k=0}^{2(N_g-1)} k^2 p_{x-y}(k) = \frac{1}{\tilde{N}_g}(m_1(d\nabla)^2 + m_2(-m_1\nabla)^2) = \nabla^2 d(\tilde{N}_g + 1 - d).\tag{16}$$

To infer the asymptotic scaling law exponents from the exact formula of DS given by Eq. (16), one drops the integer part function from $\tilde{N}_g$ and uses an approximate formula $\tilde{f}_{10} \approx \nabla d(N_g - 1) + (1 - d)\nabla^2 \approx \nabla dN_g$, which suggests the scaling law

$$f_{10} \propto N_g |d| \nabla.$$

The theoretically predicted linear scaling with $N_g$ is confirmed by numerical simulations shown in Fig. 4C, for a fixed $|d| = 1$ pixel and slopes that increase linearly with the gradient intensity $\nabla$.

The scaling of experimental $f_{10}$ with the displacement vector $d$ exhibits a linear dependence with a slope proportional to the gradient $\nabla$. Additionally, the plot of the theoretical prediction from Eq. (16) shows some deviation from linearity for large gradients. This is expected because $\tilde{f}_{10}$ neglects the contribution of the term $(1 - d)\nabla^2$ compared to $\nabla d(N_g - 1)$. However, the contribution of the neglected term increases quadratically with the gradient $\nabla$ and could become significant for images with large gradients (see Table 1).

## Entropy

The Haralick features discussed thus far are derived from different moments of the marginal distribution of either the difference intensity (see Eq. (8)) or the sum intensity (see Eq. (9)). In contrast, entropy employs a logarithmic scale to compute features from the GLCM. The definition of the entropy feature is:

$$f_9 = -\sum_{i=0}^{N_g-1} \sum_{j=0}^{N_g-1} p(i,j) \log(p(i,j)).\tag{17}$$

Entropy reaches its maximum value when a probability distribution is uniform (entirely random texture) and its minimum value of 0 when all grayscale values in the image are the same. If the entropy $f_9$ is defined using the base-2 logarithm $\log_2()$, then $f_9$ is measured in bits. While we only examined the entropy feature $f_9$, employing the marginal distributions of pixel intensity sums from Eq. (9) along with the detailed calculation examples for the SA and SV features, one can easily deduce the scaling law of SE defined by:

$$f_8 = -\sum_{k=0}^{2(N_g-1)} p_{x+y}(k) \log(p_{x+y}(k)). \tag{18}$$

Similarly, by using the marginal distribution of pixel intensity difference from Eq. (8) and the detailed calculation examples provided for the DA feature, one can easily derive the scaling law of DE defined by:

$$f_{11} = -\sum_{k=0}^{N_g-1} p_{x-y}(k) \log(p_{x-y}(k)). \tag{19}$$

Calculating entropy is straightforward because all GLCM entries carry equal weight, leading to:

$$f_9 = -\sum_{i=0}^{N_g-1}\sum_{j=0}^{N_g-1} p(i,j) \log(p(i,j)) = \log(\tilde{N}_g). \tag{20}$$

As seen from the numerical simulation results presented in Fig. 4D, the theoretical scaling law derived from Eq. (20) captures the general logarithmic trend of the entropy. However, it slightly underestimates it (see Table 1). Numerical simulations illustrated in Fig. 5D confirm that Entropy feature $f_9$ is independent of the magnitude of the displacement vector, as predicted by Eq. (20), and also slightly underestimates the actual values. The discrepancy arises from an offset constant $\varepsilon$ used in estimating the entropy from images where $\log(p(i,j) + \varepsilon)$ was employed instead of $\log(p(i,j))$ to prevent the entropy singularity for sparse GLCM.

# DISCUSSIONS AND CONCLUSION

Haralick's features are widely used in data dimensionality reduction and ML algorithms for image processing in a wide range of practical applications such as MRI (*Brynolfsson et al., 2017*) and CT scan image processing (*Cao et al., 2022*; *Chen et al., 2021*; *Park et al., 2020*; *Shafiq-ul Hassan et al., 2017*, *2018*; *Tharmaseelan et al., 2022*), cancer detection (*Faust et al., 2018*; *Cook et al., 2013*; *Permuth et al., 2016*; *Soufi, Arimura & Nagami, 2018*), liver disease (*Acharya et al., 2012*, *2016*; *Raghesh Krishnan & Sudhakar, 2013*) and mammographic masses classification (*Midya et al., 2017*), colon lesions (*Song et al., 2014*), prostatic devices for disable people (*Alshehri et al., 2024*), detection of violent crowd (*Lloyd et al., 2017*), image forensic (*Kumar, Pandey & Mishra, 2024*), malware detection (*Ahmed, Hammad & Jamil, 2024*; *Karanja, Masupe & Jeffrey, 2020*), human face detection (*Jun, Choi & Kim, 2013*), computer network intrusion detection (*Baldini, Hernandez Ramos & Amerini, 2021*). However, their interpretation poses challenges since

they are second-order statistics that depend in a complicated and nonlinear manner on image characteristics such as the number of gray level quantization $N_g$ and and the intensity of image gradients $\nabla$ and the selected displacement vector $d = (\Delta x, \Delta y)$ between adjacent pixels through the image. This study focused on extracting meaningful analytic expressions and deriving asymptotic scaling laws from Haralick's features for synthetic images containing only linear gradients. We focused on linear gradients for several reasons: (a) The human visual system efficiently decomposes and analyzes natural scenes using orthogonal gradients (*Jagadeesh & Gardner, 2022*; *Barten, 1999*; *Bracci & Op de Beeck, 2023*; *Cheng, Chen & Dilks, 2023*; *Henderson, Tarr & Wehbe, 2023*), (b) Efficient computer vision algorithms leverage gradient spectral priors to extract image features (*Gong & Sbalzarini, 2016*; *Zheng et al., 2022*), (c) In 2D natural scene images, orthogonal gradients are uncorrelated (*Gong & Sbalzarini, 2016*), and (d) The entries of the GLCM serve as natural measures of image gradients. For instance, $p_d(i, j)$ is the gradient intensity $(j - i)/|d|$ in a given image along the displacement vector $d = (\Delta x, \Delta y)$. We demonstrated that the GLCM for any linear gradient has nonzero entries solely along the two lines parallel to its principal axis diagonal shown in Fig. 3. We found that for any GLCM associated with an image gradient, the total number of entries is $\tilde{N}_g$ given by Eq. (5). The two lines parallel to the primary diagonal in Fig. 3 represent the gray level differences: (1) $k = j - i = d \cdot \nabla$, with $m_1 = \tilde{N}_g - |d|$ entries (see Eq. (6)) and (2) $k = j - i = -(1 + m_1)\nabla$, with $|d|$ entries (see Eq. (7)). Due to the GLCM symmetry for linear gradients, we derived explicit analytical expressions for the marginal probabilities $p_{x-y}(i)$ and $p_{x+y}(i)$ that are used to compute some of Haralick's features. To our knowledge, this is the only study that derived explicit mathematical expressions of Haralick's features in terms of the number of gray level quantization $N_g$, the magnitude of the linear gradient $\nabla$ present in the image, and the displacement vector $d$ used for calculating the GLCM of the image.

We found that the analytic formula for the SA $f_6$ in Eq. (10) scales linearly with the number of gray levels $N_g$ in the image and is independent of both the image gradient $\nabla$ and displacement vector $d$. The numerically estimated dependence of $f_6$ on $N_g$ shown in Fig. 4A confirms the theoretical predictions. Similarly, numerical simulations confirm that $f_6$ is independent of the magnitude image gradient $\nabla$ and the vertical displacement vector $d$ as shown in Fig. 5A.

The theoretical formula for the SV in Eq. (12) shows the asymptotic scaling law as $f_7 \propto N_g^2 |d| \nabla$. As predicted theoretically, SV increases quadratically with $N_g$, which was confirmed numerically (see Fig. 4B). The analytically predicted SV increases linearly with $d\nabla$, which was numerically confirmed in Fig. 5B, which shows that the slope of the SV *vs d* increases proportional to the gradient intensity $\nabla$.

We also predicted analytically that the DV features given by Eq. (16) has a scaling law $f_{10} \propto N_g \nabla |d|$. Our numerical simulations confirmed that SD increases linearly with $N_g$, with a slope that itself increases linearly with the image gradient $\nabla$, as shown in Fig. 4C. For a fixed $N_g = 256$, the SV increases linearly with the magnitude of the displacement vector ($|d|$), with a slope proportional to $\nabla$ (see Fig. 5C).

As we predicted theoretically, the entropy scales logarithmically with $N_g$ and $\nabla$ and is independent of $|d|$, *i.e.*, $f_9 \propto \tilde{N}_g$.

We provided a detailed derivation of exact analytic formulas and asymptotic scaling laws for the four Haralick features associated with vertical image gradients.

Since natural scenes can be decomposed into orthogonal and uncorrelated gradients (*Gong & Sbalzarini, 2016*), our derivations can be extended to a multidimensional gradient-based Haralick feature space. In our synthetic images, we introduced a single gradient along the vertical direction ($\nabla_y = \nabla$) while setting the horizontal gradient to zero ($\nabla_x = 0$) as shown in Fig. 3. This design simplified the identification of general GLCM symmetries induced by the gradient, as described in "Methods". However, our derived formulas remain valid because, even in natural scenes, orthogonal image gradients are uncorrelated.

To generalize our findings, the scalar gradient $\nabla$ must be replaced with the gradient vector ($\nabla_x, \nabla_y$) for 2D images. The analytical formulas we derived for Haralick's features can be used to estimate image gradients from measured features. Another application involves deriving consistent normalization factors for Haralick features. Comparing the values of Haralick features across datasets from different scanners with varying resolutions is challenging and different empirical normalizations algorithms achieved only limited success (*Clausi, 2002*; *Lofstedt et al., 2019*; *Shafiq-ul Hassan et al., 2017*, *2018*). Thus, identifying suitable normalization factors that render Haralick features invariant to the number of gray levels or the quantization scheme is crucial among other fields in radionics.

We demonstrated that the SA feature in Eq. (10) should be normalized by $N_g$ to ensure asymptotic independence from the quantization scheme. This normalization allows for the consistent comparison of the Haralick SA feature across images obtained at different resolutions and with various imaging devices. Similarly, we analytically proved that the SV feature in Eq. (12) should be normalized by $N_g^2$ to achieve invariance to the image quantization scheme. Unlike empirical trial-and-error approaches, our normalization factors are rigorously derived based on the symmetries of the GLCM, ensuring mathematical consistency and robustness.

## ACKNOWLEDGEMENTS

The content is solely the responsibility of the authors and does not necessarily represent the official views of the National Institutes of Health.

### Funding

A South Carolina Space Grant Consortium Palmetto Academy grant, a Research and Development grant from the College of Charleston, and the National Institute of General Medical Sciences of the National Institutes of Health under Award Number P20GM103499 supported this research. The funders had no role in study design, data collection and analysis, decision to publish, or preparation of the manuscript.

## Grant Disclosures

The following grant information was disclosed by the authors:

South Carolina Space Grant Consortium Palmetto Academy.

College of Charleston.

National Institute of General Medical Sciences of the National Institutes of Health: P20GM103499.

## Competing Interests

The authors declare that they have no competing interests.

## Author Contributions

- Sorinel A. Oprisan conceived and designed the experiments, performed the experiments, analyzed the data, performed the computation work, prepared figures and/or tables, authored or reviewed drafts of the article, and approved the final draft.
- Ana Oprisan analyzed the data, prepared figures and/or tables, authored or reviewed drafts of the article, and approved the final draft.

## Data Availability

This is a theoretical study. The code is available in the Supplemental Files.

## Supplemental Information

Supplemental information for this article can be found online at http://dx.doi.org/10.7717/peerj-cs.2856#supplemental-information.

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
