# Peer review of "Scaling laws for Haralick texture features of linear gradients"

_PeerJ Computer Science, doi:10.7717/peerj-cs.2856_

## Round 0.1 · original submission · Major Revisions

Please read the comments carefully and address the issues accordingly. Then your work will be evaluated again.

Reviewer 1 ·

Basic reporting

The article adheres to academic standards is clearly written and is well-structured. With relevant references and well-structured parts, it offers a solid theoretical basis. Even if the pictures are instructive some of them should be made clearer, especially for readers who are not experienced with texture analysis. In the introduction the emphasis on medical imaging stands in stark contrast to the study's higher mathematical focus.

Experimental design

The study rigorously derives scaling laws for four Haralick texture features from synthetic gradient images. The controlled approach is appropriate but its limit to linear gradients may not fully reflect the real-world textures of the land. Numerical validation using MATLAB is reasonably reasonable, but further tests with real images would strengthen the findings.

Validity of the findings

Numerical simulations and theoretical predictions agree well, however little variations in entropy point to the need for improvement. Although the suggested scaling rules have potential uses in medical imaging and machine learning, their practicality has to be verified through validation on a variety of datasets.

Additional comments

1. It is important to talk about broader applications outside of medical imaging.
2. More thought should be given to the effects of periodic boundary conditions.
3. Figures may use a clearer explanation, particularly Fig. 3.
4. It is important to consider computational efficiency in comparison to conventional techniques.
5. Complex textures and non-linear gradients should be explored in future development.

Cite this review as

Reviewer 2 ·

Basic reporting

1. The introduction should be strengthened by briefly mentioning the problem or gap this framework addresses. For instance, why is scaling behavior important for texture analysis? Does it improve feature consistency or result in more robust classification in medical imaging? Providing a hint of the underlying problem or need for this framework would ground the study for the reader.
2. The term “synthetic gradient images” could also be explained more briefly in the research methods. Are these synthetic gradients meant to simulate real-world images, or are they used for controlled testing? A clearer connection between synthetic gradient images and the broader applicability of this work might be helpful.
3. Some of references are quite old, should be updated with 5 years latest references.
4. Need more detail and specific about hypotheses which can make the research direction.
5. All the article structure, figures, and tables are satisfied enough, also all the formal results already clear in all terms and theorems.

Experimental design

The experimental design should be contained as follows:
1. Details on the creation of synthetic gradient images: Explain how the images were generated, what variations were applied to Ng, d, and ∇, and how these variables were systematically tested.
2. Clarification of the simulation setup: Provide more information about how numerical simulations were conducted, what software or models were used, and what specific comparisons were made between theory and simulation.
3. Validation process: Briefly mention how the theoretical predictions were validated (e.g., through comparison with real-world data, alternative methods, or performance metrics like classification accuracy).
Metrics for success: Specify how the performance of the texture features and the effectiveness of the normalization were assessed.

By adding these details, the experimental design would be clearer and more comprehensive, allowing the reader to fully grasp how the study was conducted and how the results were validated.

Validity of the findings

For the the validation of the findings, the discussion should explain about:
1. It could be useful to clarify what metrics were used to evaluate the success of the normalization. Did you use classification accuracy, feature stability, or computational efficiency as metrics? Indicating how success is measured will strengthen the explanation of your experimental design.
2. The manuscript suggests that the experiments likely involved evaluating the performance of texture features before and after normalization and could much more clear if possibly comparing them by using across datasets.

Cite this review as

---

## Round 0.2 · accepted · Accept

Thanks a lot to the authors for their efforts to improve the work. I believe the current version can be accepted. Congrats!

Reviewer 2 ·

Basic reporting

The explanations in the response are clear enough, so there is no need for further revisions.

Experimental design

The explanations in the response are clear enough, so there is no need for further revisions.

Validity of the findings

The explanations in the response are clear enough, so there is no need for further revisions.

Additional comments

It can continue to be published.

Cite this review as